# Heterogeneous Network Switching Strategy Based on Communication Blind Area Dwell Time

**DOI:** 10.3390/s23136166

**Published:** 2023-07-05

**Authors:** Cheng Zhang, Yanfeng Tang, Xiuzhuo Wang, Yan Zhang, Xiuyang Li

**Affiliations:** School of Electronics and Communication Engineering, Changchun University of Science and Technology, Changchun 130022, China; 2021100795@mails.cust.edu.cn (C.Z.);

**Keywords:** visible light communication, heterogeneous networking, ACO-OFDM, communication blind area, horizontal–vertical collaborative handover

## Abstract

The limitation of indoor visible light coverage and the attenuation of its signal when propagating in line-of-sight has seriously affected the stable communication of receiving devices when users move randomly and also aggravated the power consumption of visible light networking systems. According to the above situation, on the basis of the heterogeneous networking of visible light communication (VLC) and RF communication integration, this article proposes a horizontal–vertical collaborative handover strategy based on the communication blind area dwell time (CBD-HVHO). Combining asymmetrically clipped optical orthogonal frequency division multiplexing (ACO-OFDM) technology with networking handover technology, ACO-OFDM is used to determine the indoor communication blind area by calculating the bit error rate (BER) value at the signal receiver while reducing the multipath interference generated by visible light signals during channel transmission. To achieve this, set the communication blind channel interruption time as the threshold time, compare the communication blind area dwell time with the threshold time, and finally combine the horizontal and vertical collaborative handover strategies based on the communication blind area dwell time. The simulation results show that the handover probability is 0.009, the average number of handovers is 1.006, and the average network throughput is 195.2826 Mbps. Compared with the previously proposed immediate vertical handover (I-VHO) scheme and the dwell vertical handover (D-VHO) scheme, the communication stability is significantly improved, and the power consumption of the network system is reduced to a certain extent.

## 1. Introduction

With humans’ constant research and experiments on information technology, wireless communication technology has been effectively developed, and a new communication method, VLC, through ongoing experiments and improvements, has gradually started to replace traditional wireless communication technology [1,2]. Compared with traditional RF communication, VLC has the advantages of low carbon and environmental protection, abundant bandwidth resources, and high-security performance. In order to overcome the problems of unstable reception of signals from user-receiving equipment and high power consumption of communication systems under visible light communication, research on network handover technology is necessary.

Visible light communication heterogeneous networking technology has attracted a lot of attention in today’s academic circles due to its unique communication advantages [3]. In visible light communication heterogeneous networking technology, horizontal handover (HHO) is the switching between multiple visible light networks, and vertical handover (VHO) is the switching between Wi-Fi and visible light networks [4]. Indoor daily office areas generally adopt integrated networks of VLC and Wi-Fi, and because Wi-Fi can cover the whole area, the VHO scheme is proposed to be progressively popular in office areas [5]. Hou et al. [6] came up with a dwell vertical handover (D-VHO) scheme where the vertical handover controller could predict the link recovery time by waiting for a fixed duration. However, the drawback is that the vertical handover is executed only when the optical link remains blocked at the end of the waiting time. Bao et al. [7] proposed a channel adaptive residence vertical handover (CAD-VHO) scheme, which could use the channel blocking rate and recovery rate to determine the adaptive residence time and then determine whether to perform vertical handover. Nevertheless, the shortcoming is that it does not take real-time application operations into account. Subsequently, their team members, Andrews et al. [8], proposed the technique of hybrid application-aware VHO (HA-VHO), which took real-time applications into consideration. However, they only analyzed the communication in the overlapping region of visible light coverage. Wu [9] made use of mines as an application scenario to learn the heterogeneous networks and proposed a received signal strength (RSS) horizontal handover scheme on the basis of fast Fourier transformation (FFT) fading detection for coal mine working faces and a dynamic residence time vertical handover scheme for roadways, while the communication between coal mine working faces and roadway handover areas failed to be considered. Wang et al. [10] proposed a heterogeneous network handover strategy that depended on the quality of service of users, using network dwell time as the handover mechanism and combining vertical handover algorithms on top of horizontal handover, but also failed to analyze the communication in the region of severe fading of visible light signals.

In summary, most of the current research on visible heterogeneous networking handover technology fails to take into account all visible signal attenuation areas, which are collectively referred to as communication blind areas in this article. ACO-OFDM technology is used to process visible signals and determine the communication blind zone by calculating the BER at the receiver side, which improves the recognition of the communication blind area, followed by considering the communication stability in the communication blind area, focusing on the dwell time of the user receiving device in the communication blind area, using the channel interruption time as the threshold time, and developing horizontal–vertical collaborative switching by comparing the dwell time with the threshold time. By comparing the dwell time with the threshold time, the horizontal–vertical cooperative handover algorithm is developed, which improves the communication stability of the user receiving equipment and reduces the power consumption of the networking system.

The remainder of this article is presented as follows: Section 2 discusses the basic theory of visible light communication and the basic framework model of ACO-OFDM. Section 3 shows the visible light communication network light source layout design, ACO-OFDM channel processing, and the flow of the horizontal–vertical collaborative handover algorithm based on communication blind area residence time. Section 4 evaluates the performance of the horizontal–vertical collaborative handover algorithm based on the communication blind dwell time, which assesses the average network throughput, the average number of handovers, and the handover failure probability of the networking system. Section 5 concludes the article.

## 2. Methodology

### 2.1. Basic Theory of VLC

Line of sight (LOS) and non-line of sight (NLOS) are the two main types of indoor optical communication links. The experiment results demonstrate that the received power of indoor visible direct link accounts for 93.03% of the total received power, and the received power of primary reflection accounts for 5.53% of the total received power [11]. In order to reduce the errors generated during the process of calculating the BER, we only adopt LOS in this paper.

The light-emitting diode (LED) light source distribution obeys the Lambert distribution, and the specific Lambert model can be expressed as
(1)$$R\left(\theta \right)=\frac{m+1}{2\pi }\cos^{m}\left(\theta \right)$$
where θ indicates the radiation angle of the LED light source, and *m* is the number of radiation modes of the light source, which is related to the half-power angle of the light source.

In the case of LOS, the channel direct current (DC) gain is calculated by
(2)$$H\left(0\right)=\left\{\begin{array}{l} \frac{m+1}{2\pi d^{2}}A\cos^{m}\left(\theta \right)T_{s}\left(\psi \right)g\left(\psi \right)\cos\left(\psi \right),0\leq \psi \leq \psi_{c}\\ 0,\psi >\psi_{c} \end{array}\right.$$
where *A* is the sensing area, $T_{S}\left(\psi \right)$ is the optical filter DC gain, $\psi_{c}$ represents the half field of view of the receiver, and *d* represents the position from the light source. The $g\left(\psi \right)$ is the gain of the concentrator, which can be presented in the following equation:(3)$$g\left(\psi \right)=\left\{\begin{array}{l} \frac{n}{\sin^{2}\psi_{c}},0\leq \psi \leq \psi_{c}\\ 0,\psi >\psi_{c} \end{array}\right.$$

The received power under the conditions of the LOS is [12,13]:(4)$$P_{R}=P_{s}H\left(0\right)$$
where *P_S_* represents the transmit power, and *P_R_* represents the received power under the circumstance of LOS.

In the channel of indoor visible light transmission, Gaussian white noise is employed. There are two main types of noise: scattered particle noise and thermal noise. The scattered particle noise can be expressed as:(5)$$\delta_{shot}^{2}=2qR\left(P_{R\left(signal\right)}+P_{R\left(ISI\right)}\right)B+2qI_{bg}I_{2}B$$
where *q* is the quantity of electric charge, *P_R_*_(*signal*)_ is the signal received power, *B* is the noise bandwidth, *P_R_*_(*ISI*)_ is the inter-code interference power, *I_bg_* is the dark current, and *I*_2_ is the noise bandwidth factor. The thermal noise can be expressed as
(6)$$\delta_{thermal}^{2}=\frac{8\pi kT_{k}}{G}C_{pd}AI_{2}B^{2}+\frac{16\pi^{2}kT_{k}\Gamma }{g_{m}}C_{pd}^{2}A^{2}I_{3}B^{3}$$
where *k* is the Boltzmann constant, *T_k_* is the absolute temperature, *C_pd_* is the fixed capacitance per unit area of the photodetector, *G* is the open-loop voltage gain, Γ is the channel noise coefficient, *g_m_* is the transconductance coefficient, and *I*_2_, *I*_3_ are both noise bandwidth coefficients. The signal-to-noise ratio (*SNR*) can be expressed as [14]:(7)$$SNR=\frac{R^{2}P_{R(signal)}^{2}}{\delta_{shot}^{2}+\delta_{thermal}^{2}}$$

### 2.2. The Technology of ACO-OFDM

A significant problem is multipath interference during the channel transmission by the indoor visible light signal. The orthogonal frequency division multiplexing (OFDM) technology is widely studied and applied and has the benefits of high-frequency utilization and strong resistance to multipath interference. In order to satisfy the requirement of non-negative real signals and improve the power efficiency of the system at the same time in the situation of introducing OFDM into visible optical communication, researchers came up with a single-polarity real-valued scheme called the ACO-OFDM technique [15].

As shown above, in Figure 1, the data input at the transmitter side and the data output at the receiver side are both binary serial data streams. Due to the limitation that light signals can only propagate in a positive direction, the limiting operation is performed to output a positive real number signal, and the cyclic prefix (CP) is added to eliminate inter-code crosstalk during the channel transmission.

## 3. VLC Heterogeneous Networking Technology Research

### 3.1. Visible Light Source Layout

This program built a square room model with the size of 6 m × 6 m × 3 m. The layout design used to cover the radius of 1 m and $\sqrt{2}$ m LED light source, respectively. Among them are the 1 m LED light source with the distribution of a diamond-shaped layout and the $\sqrt{2}$ m LED light source with the distribution of an x-shaped layout. Laying out a Wi-Fi RF communication is laid out in the middle area, where the overall arrangement of the light source is depicted in Figure 2:

Figure 3 shows the light intensity distribution depending on the visible light source layout, which has a maximum light intensity of 1126.59*lx* and a minimum value of 215.63*lx*. Moreover, the average light intensity is 367.22*lx*, which meets the international standard for indoor lighting [16]. Therefore, according to the figures above, the layout of the visible light source designed in this scheme can provide communication for the user receiving the equipment but also take the lighting into account.

### 3.2. ACO-OFDM Channel Processing

In the visible light channel transmission, unlike the traditional method of introducing SNR, we introduce spatial SNR on the basis of the layout of the visible light source in this article (Figure 4).

After the operations of quadrature amplitude modulation (QAM), Hermitian symmetry, inverse fast Fourier transformation (IFFT) transformation, adding CP, and limiting operations in the transmitter, the bit stream can be transmitted in an additive white Gaussian noise (AWGN) [17] channel through digital-to-analog conversion. Analogously, the receiver side is equivalent to the inverse process of the transmitter side. Finally, we can obtain the BER by calculating the error between the output and the input bit stream at the receiver side and converting the BER into the indoor spatial distribution (Figure 5).

### 3.3. CBDHVHO Scheme

In this article, we utilize Wi-Fi for uplink data transmission. Meanwhile, we employ both VLC and Wi-Fi for downline data transmission. Furthermore, we analyze three situations of communication blind areas based on the light source layout: (1) The visible light coverage overlapping area. (2) The visible light coverage tangent area. (3) The visible light has no coverage area. The flow chart of the specific handover strategy is shown in Figure 6.

In the above flowchart, the communication blind area dwell time *Ts* refers to the time from the mobile receiving device entering the communication blind zone to the time of leaving. The carrier-to-noise ratio is normal, and the case of BER > $10^{-4}$ is chosen as the communication blind area in this paper. *T_threshold_* refers to the threshold time, which follows a random distribution in the interval [*T_res_*, *T_max_*]. *T_res_* is the time limit of the response time for the user’s thought stream to remain uninterrupted [18], and *T_max_* is the maximum waiting time for the visible channel interruption. When the *T_S_* is greater than *T_threshold_*, the user receiving equipment in the communication blind area will be affected by the channel interruption. In order to maintain normal communication, the vertical handover is accessed at this time, and in order to prevent the switching handover cost from being too high, the execution of the vertical handover will be maintained until the end of the mobile process. When the *T_S_* is less than the *T_threshold_*, the user’s receiving device is basically not affected by the channel response interruption. The implementation of horizontal handover can ensure high-quality user communication.

## 4. Simulation Results

This experiment uses a room layout of 6 m × 6 m × 3 m, where the coordinates of LED light sources with a covering radius of 1 m are (1, 3), (3, 1), (3, 5), and (5, 3). The coordinates of LED light sources with a covering radius of $\sqrt{2}$ m are (1, 1), (1, 5), (3, 3), (5, 1), and (5, 5). The user-receiving device moves randomly in the speed range of 0.3 to 0.7 m/s. The time that the user receiving device moves at a value higher than the normal communication BER is the communication blind spot dwell time, and the time required for network handover is the handover delay time, as shown in Table 1.

In the whole visible optical communication networking system, in order to highlight the communication stability of the CBDHVHO solution, the BER and the probability of handover failure are first used as performance indicators. The BER is a measure of the accuracy of data transmission within a specified period of time, and the size of the handover failure probability affects the user’s voice call and Internet experience. The average number of handovers reflects the signaling cost of the visible optical communication system, and the lower number of handovers avoids the ping-pong effect it generates. Finally, the average network throughput is used as a communication quality indicator, and high data throughput can support functions such as high-definition video calls and livestreaming even when users are mobile. Also, the I-VHO scheme [19] and the D-VHO scheme are used as comparison experiments to highlight the better performance of the CBDHVHO scheme through comparison experiments.

### 4.1. Communication Stability Performance Evaluation

In this paper, the handover failure probability and BER are used as performance indicators, and the DC-biased optical orthogonal frequency division multiplexing (DCO-OFDM) [20] is used as the comparison scheme for the BER experiment. And the handover failure probability equation is as follows:(8)$$F_{HO}=\frac{\sum_{r=1}^{N_{r}}\frac{P{\left(r\right)}^{B}-P{\left(r\right)}^{B+1}}{1-P{\left(r\right)}^{B+1}}}{N_{r}}$$
where *F_HO_* is the handover failure probability of the specified scheme, *P*(*r*) is the utilization of the RF communication uplink server after *r* iterations, and *B* is the maximum queue length of the RF communication uplink.

#### 4.1.1. Performance Analysis for Handover Failure Probability

From Figure 7, the I-VHO scheme has the highest handover failure probability of 0.018, which is caused by frequent handovers. The D-VHO_t=0.5_ scheme and D-VHO_t=1_ scheme have a handover failure probability of 0.016 and 0.013, respectively, which is because the scheme does not perform handover during partial dwell time, and the CBDHVHO scheme has a handover failure probability of 0.009 due to its lower handover cost and seamless connection. The handover failure probability is 0.009 because the handover failure probability is closely related to the number of handovers; more handovers will result in more delay time and more dwell packets, so the handover failure probability will increase.

#### 4.1.2. Performance Analysis for BER

The magnitude of the BER is related to the SNR, and it can be seen from Figure 8 that the BER decreases as the SNR increases. When the SNR is almost zero, the BER of the DCO-OFDM scheme is 0.3096, and the BER of the ACO-OFDM scheme is 0.4346. As the SNR increases to 20 dB, the BER of DCO-OFDM is 0.1605. While the BER of the ACO-OFDM scheme is $5.722\times 10^{-6}$. The lower BER also means that the received signal error is smaller and the communication stability is better. Therefore, it can be seen that ACO-OFDM is more resistant to multipath interference and has a great improvement in communication performance.

### 4.2. Heterogeneous Networking System Power Performance Evaluation

The average number of handovers can be calculated by
(9)$$A_{HO}=\frac{\sum_{r=1}^{N_{r}}N_{HO}\left(r\right)}{N_{r}}$$
where *A_HO_* presents the average number of handovers and *N_r_* represents the number of executions. *N_HO_*(*r*) represents the number of *r*-th execution.

#### Analysis of the Average Number of Handovers for Different Programs

As Figure 9 shows the average number of handovers under the comparison of various schemes, a total of 1000 experiments were performed, where CBDHHO represents the horizontal handover of the CBDHVHO scheme, and CBDVHO represents the vertical handover of the CBDHVHO scheme. From the figure, it can be seen that the I-VHO scheme has the highest average number of handovers, which is 1.875. The reason is that the number of handovers in this scheme increases with the change in the VLC network. The D-VHO_t=0.5_ scheme sets the residence time under a VLC network to 0.5 s, which means that when the residence time of a user receiving equipment under a VLC network is lower than 0.5 s, no handover is performed, so the average number of handovers is lower, 1.424. Similarly, the average number of handovers for the D-VHO_t=1_ scheme is 1.04. The average number of handovers for the CBDHHO scheme is 1.006, but considering that the channel response interruption time in the communication blind area is very short, the CBDVHO scheme is used to assist in order to maintain continuous and stable communication with the user receiving equipment, so the total number of handovers will increase a little compared to the D-VHO_t=1_ scheme.

### 4.3. Communication Quality Performance Evaluation

The average network throughput is calculated as follows:(10)$$A_{th}=\frac{\sum_{r=1}^{N_{r}}\sum_{i=1}^{N_{i}\left(r\right)}O\left(i,r\right)\times \left[T_{i}\left(r\right)-T_{d}\left(r\right)\right]}{\sum_{r=1}^{N_{r}}\sum_{i=1}^{N_{i}\left(r\right)}T_{i}\left(r\right)}$$
where *A_th_* is the average throughput, *T_i_*(*r*) represents the network dwell time when executing the *r*-th time, and *T_d_*(*r*) represents the network handover delay time when executing the *r*-th time.

#### 4.3.1. Analysis of the Average Network Throughput of Different Solutions

The average network throughput under the comparison of various schemes is shown in Figure 10. The I-VHO scheme has the lowest average network throughput of 142.3349 Mbps due to the ping-pong effect caused by the frequent handover. It has an average network throughput of 159.3888 Mbps. The CBDHVHO scheme takes into account the continuous and stable communication of user-receiving devices, and the handover technique achieves a seamless connection, so the average network throughput is the highest at 195.2826 Mbps.

#### 4.3.2. Analysis of the Impact of Different Velocities on the Average Network Throughput

Figure 11 shows that different mobile velocities have different effects on the average network throughput. The change in movement speed affects the dwell time and communication blind zone residence time under the VLC network, and as the movement speed increases, the network dwell time and communication blind zone residence time become shorter accordingly. Since the I-VHO and D-VHO solutions are vertical handover types, the average network throughput shows a decreasing trend, while the CBDHVHO solution focuses on the horizontal handover type. The shorter the communication blind zone residence time, the more horizontal handover there is, so the average network throughput shows an increasing trend. When the mobile speed increases from 0.3 m/s to 0.7 m/s, the average network throughput of the I-VHO scheme decreases from 162.4467 Mbps to 131.5774 Mbps; the D-VHO_t=0.5_ scheme decreases from 162.8908 Mbps to 137.7859 Mbps; the D-VHO_t=1_ scheme decreases from 167.0896 Mbps to 127.5167 Mbps; and the average network throughput of the CBDHVHO scheme increases from 172.4695 Mbps to 231.5928 Mbps. The above data show that the scheme has good applicability for mobile velocity.

## 5. Conclusions

The research in this article is presented in the model of a user quality of service-based switching strategy for heterogeneous networks [10], which first detects the dwell time of the user receiving device under the visible channel, sets a threshold value, and performs horizontal and vertical switching by comparing the dwell time with the threshold value. Although this model improves the communication quality, it ignores the communication in the channel fading region, so the communication stability is not strong. Based on this, this paper uses ACO-OFDM technology to reduce multipath interference in channel transmission while using the BER of the signal receiver to determine the indoor communication blind zone, which not only enhances the signal communication stability but also improves the recognition of the communication blind zone. At the same time, combined with the horizontal and vertical collaborative handover strategies based on the communication blind zone, it continues to maintain communication stability while minimizing power consumption and improving the communication quality of the networking system. In the future, it can also contribute to the research of high-precision positioning techniques in indoor non-smooth environments [21] and the development and operation of high-data-throughput real-time applications in commercial network environments [22]. In addition, the scheme is based on the data obtained in the MATLAB simulation case, and further physical platforms need to be built to perform experimental validation. At the same time, since it is a simulation environment, we only introduced the additive white Gaussian noise channel and need to further consider the Rayleigh fading channel [23], the Rician fading channel [24], etc., to simulate the real environment. Another unconsidered factor is the self-blocking of the visible channel, which means that the visible channel is blocked by moving objects and may last for some time, so further experiments are needed to improve the indoor network switching mechanism.

## Figures and Tables

**Figure 1 sensors-23-06166-f001:**
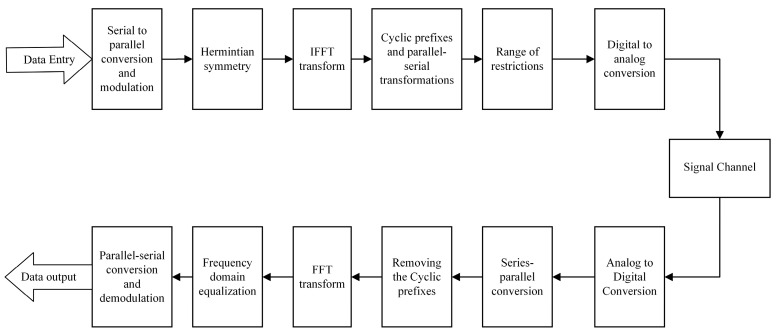
Block diagram of ACO-OFDM system.

**Figure 2 sensors-23-06166-f002:**
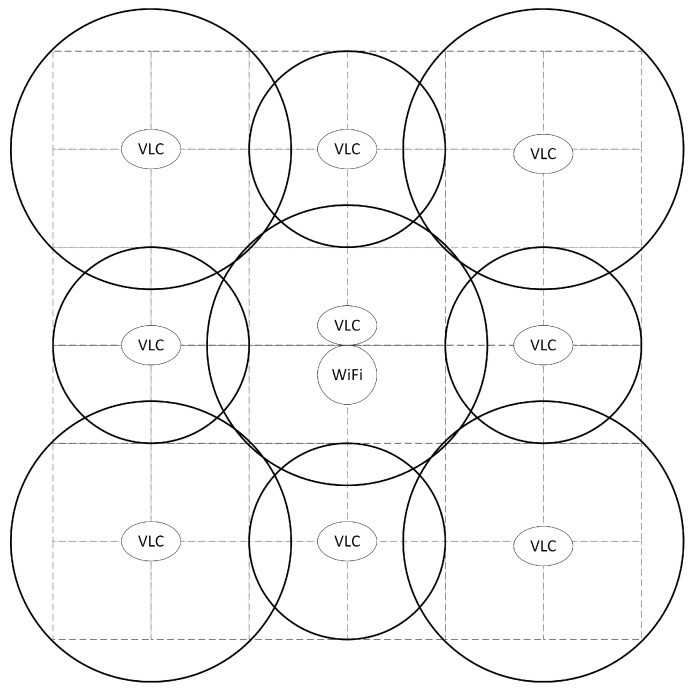
Visible light source layout.

**Figure 3 sensors-23-06166-f003:**
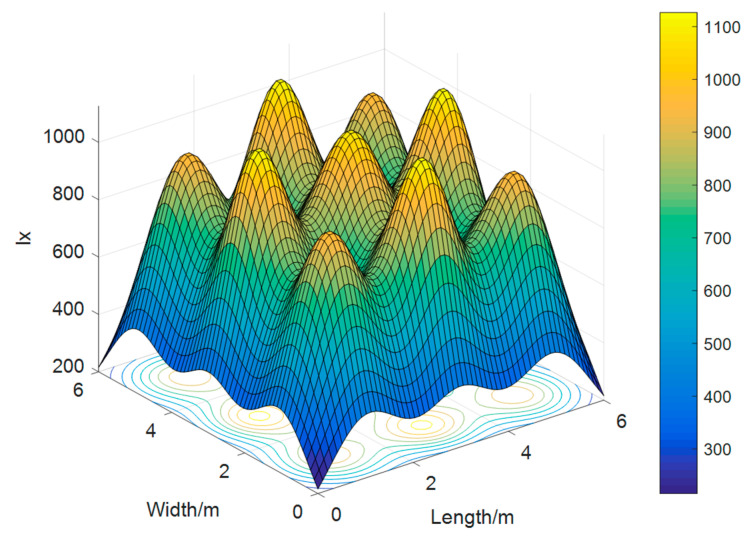
Visible light intensity distribution.

**Figure 4 sensors-23-06166-f004:**
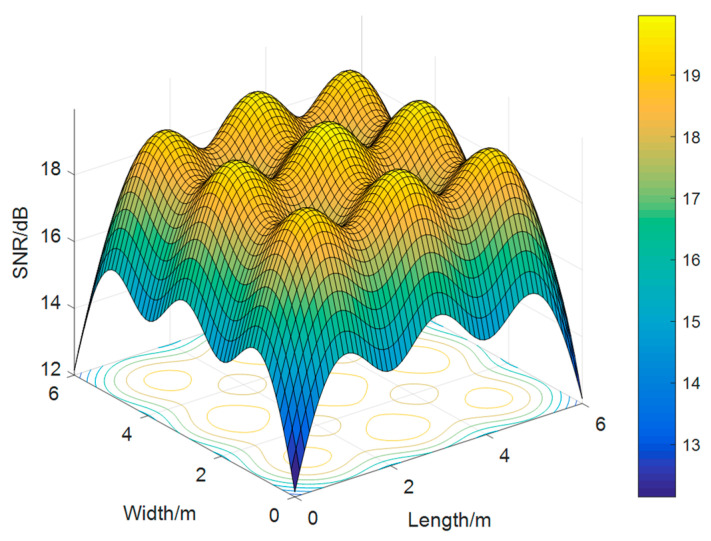
Visible light SNR distribution.

**Figure 5 sensors-23-06166-f005:**
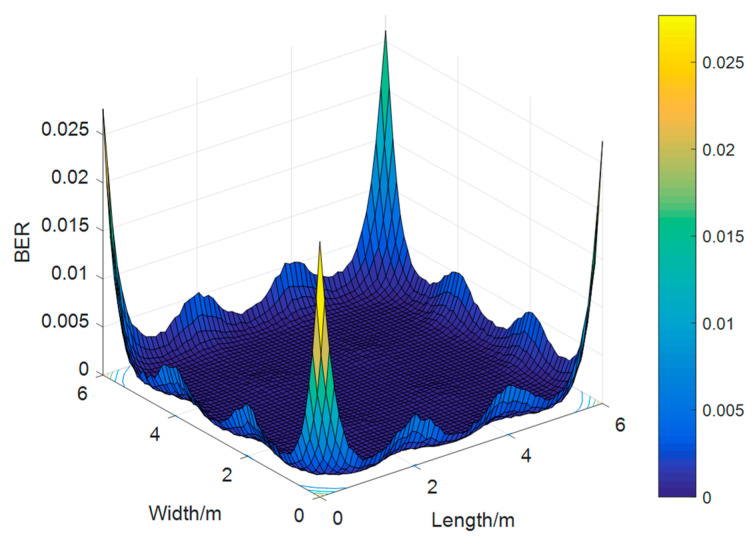
Visible BER distribution.

**Figure 6 sensors-23-06166-f006:**
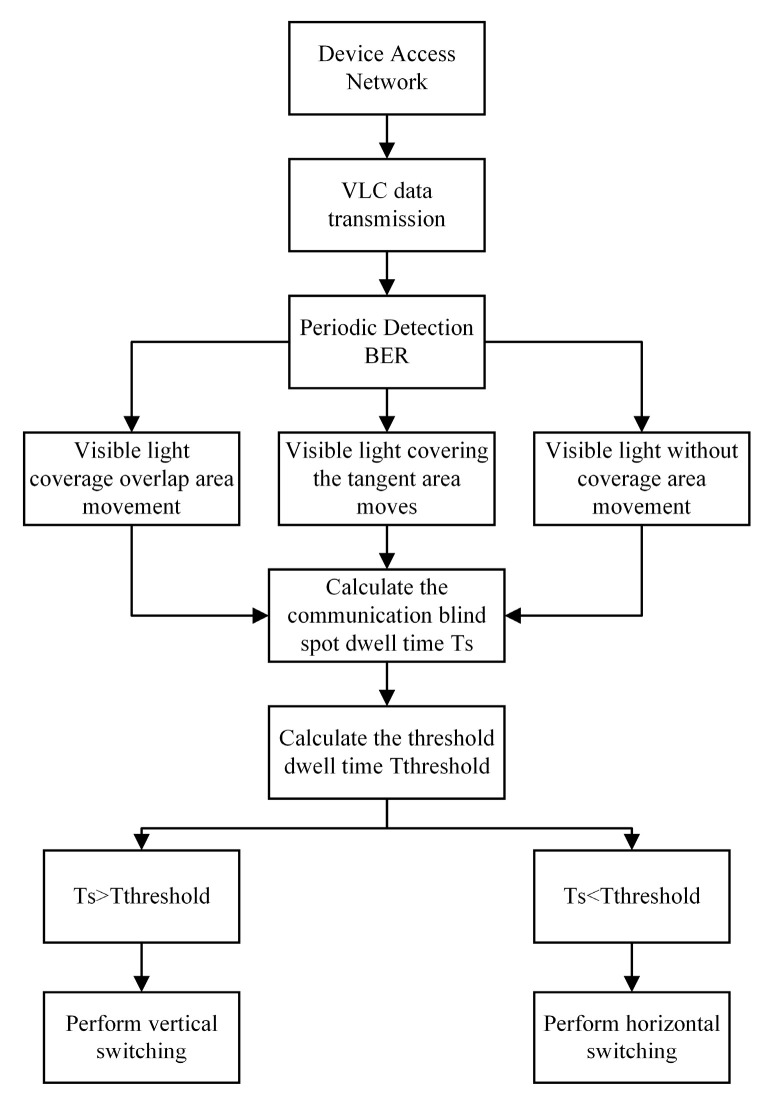
CBDHVHO scheme flowchart.

**Figure 7 sensors-23-06166-f007:**
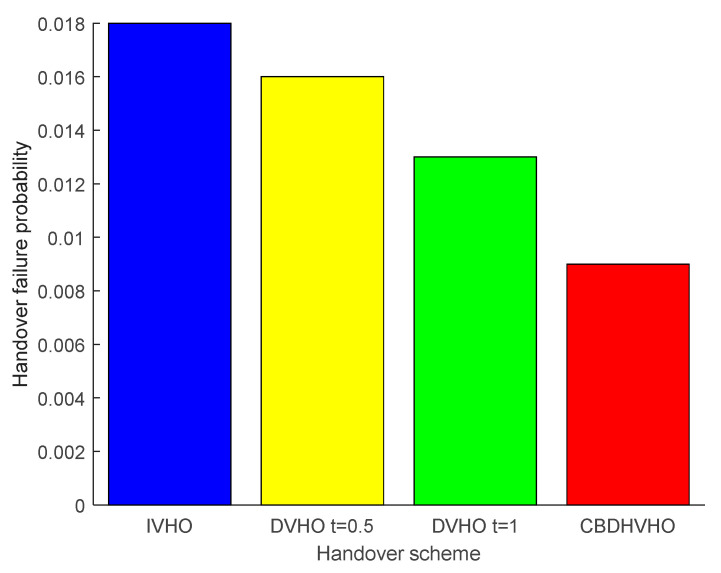
Comparison of the handover failure probabilities of the proposed I-VHO, D-VHO, and CBDHVHO schemes.

**Figure 8 sensors-23-06166-f008:**
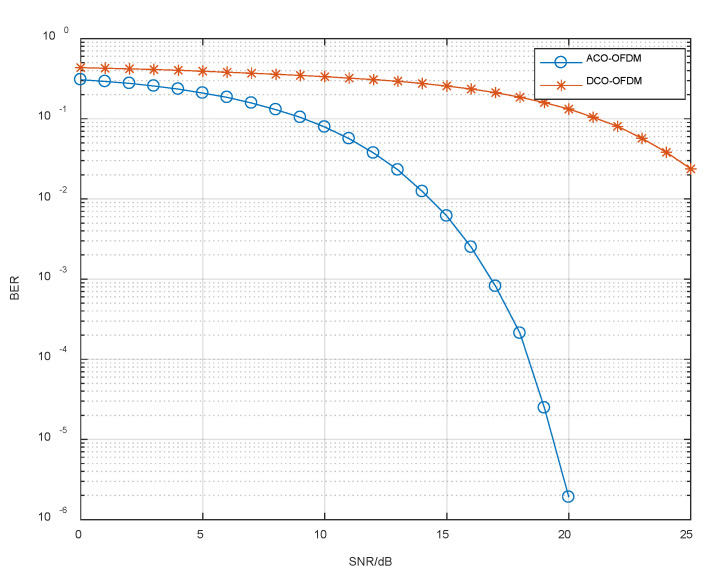
BER comparison of proposed ACO-OFDM and DCO-OFDM.

**Figure 9 sensors-23-06166-f009:**
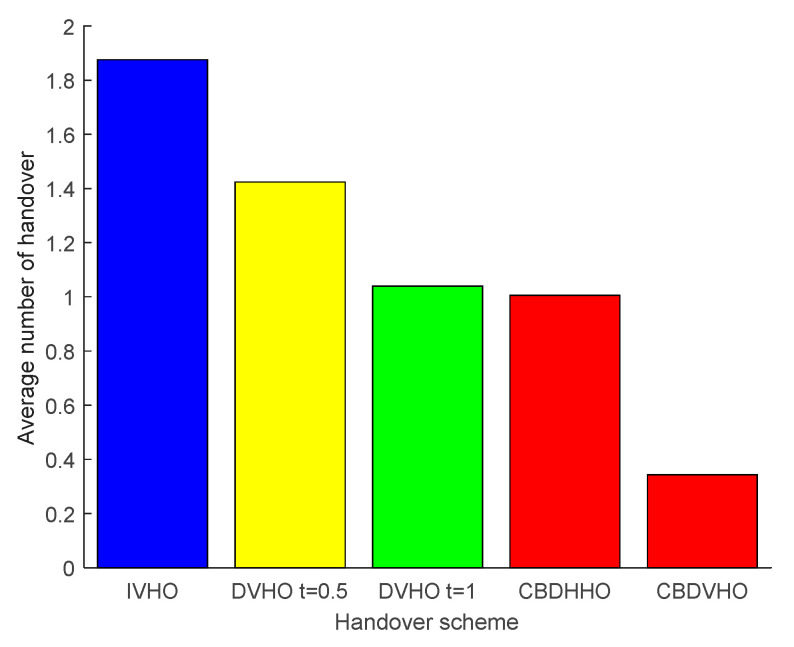
Comparison of the average number of handovers for the proposed I-VHO, D-VHO, CBDVHO, and CBDHHO schemes.

**Figure 10 sensors-23-06166-f010:**
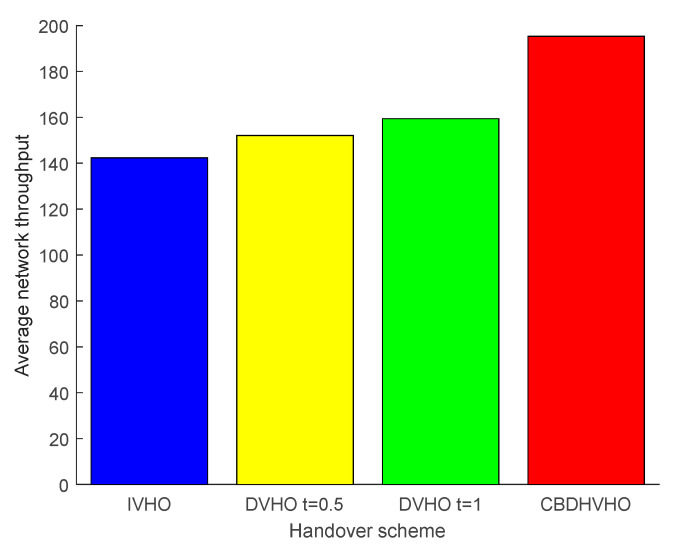
Comparison of the average network throughput of the proposed I-VHO, D-VHO, and CBDHVHO schemes.

**Figure 11 sensors-23-06166-f011:**
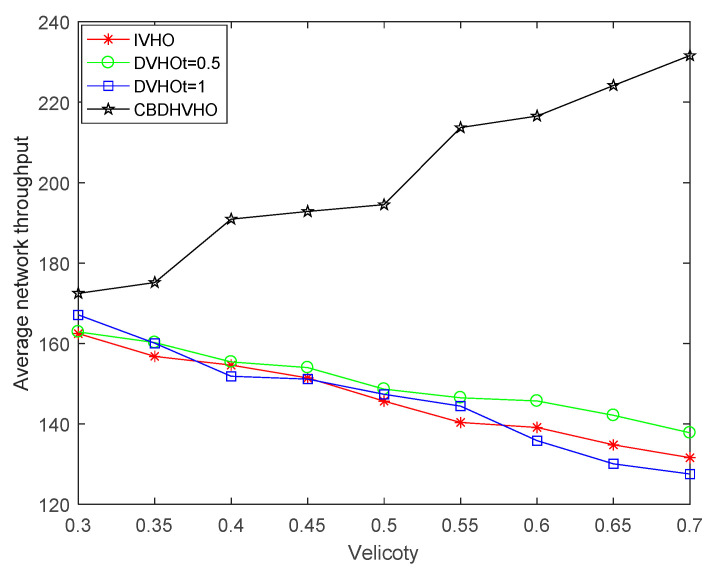
Comparison of the impact of different velocities of the proposed I-VHO, D-VHO, and CBDHVHO schemes on the average network throughput.

**Table 1 sensors-23-06166-t001:** Simulation parameters.

Parameter	Value
Room size	6 m × 6 m × 3 m
Velocity	0.3~0.7 m/s
Movement time duration	1–30 s
Throughput for VLC	400 Mbps
Throughput for Wi-Fi	100 Mbps
Number of VLC APs	9
Radius of APs	$1\;m,\;\sqrt{2}$ m
Number of iterations	1000
Number of RF APs	1
Handover delay time	0.1~1 s
Time frame for response time	1 s
Maximum waiting time	2 s
BER Scope	$10^{-7}$~0.025
Communication blind area dwell time	0~7 s

## Data Availability

Not applicable.

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
