# Peer review of "Heterogeneous Network Switching Strategy Based on Communication Blind Area Dwell Time"

_sensors, 2023, doi:10.3390/s23136166_

Round 1
Reviewer 1 Report
The authors have carried out a worthwhile piece of work.
Author Response
请参阅附件。
Reviewer 2 Report
This manuscript proposed a horizontal-vertical collaborative switching strategy based on the residency time of the communication blind area, the topic looks interesting. More details please refer to the following comments:
1) English writing improvement is strongly suggested. For example, the title should be changed, not using “Research on***”.
2) Both the motivations and contributions are not clear in Abstract and Introduction, please refine them.
3) Separate related work section should be considered to make comprehensively review for state-of-the-arts in this area.
4) High-quality figures are strongly suggested to better demonstrate both the proposed method the experimental results.
5) More evaluation metrics and state-of-the-arts should be considered to make the experimental results more sufficient.
6) Please add more references from latest three years to make the reference list more solid.
7) Separate discussion section is suggested to discuss the limitations of the proposed method, challenging issues, and future directions. For example, the authors should discuss the potential applications of the proposed method in IoT-based system. Some related papers are recommended, which are better included in the reference list: online spatiotemporal modeling for robust and lightweight device-free localization in nonstationary environments, IEEE TII, and handgest: hierarchical sensing for robust in-the-air handwriting recognition with commodity wifi devices, IEEE IoT.
English writing improvement should be considered.
Reviewer 3 Report
The contribution part needs some justification. You also need to justify why this work is needed. I can’t see such a information in this work.
The proposed model is empty. In this section, you have to talk about your suggested prototype that how it works. I can see you have only discussed relevant literature.
The experiment set-up also needed to be explain further.
What are the limitations of your model?
N/A
Round 2
Reviewer 2 Report
The quality of this paper has been enhanced, I have no more comments.
The quality of this paper has been enhanced, I have no more comments.
Reviewer 3 Report
I do not have further suggestions for the authors
N/A